# UNDERSTANDING NONLINEAR IMPLICIT BIAS VIA REGION COUNTS IN INPUT SPACE

## ABSTRACT

One explanation for the strong generalization ability of neural networks is implicit bias. Yet, the definition and mechanism of implicit bias in non-linear contexts remains little understood. In this work, we propose to characterize implicit bias by the count of connected regions in the input space with the same predicted label. Compared with parameter-dependent metrics (e.g., norm or normalized margin), region count can be better adapted to nonlinear, overparameterized models, because it is determined by the function mapping and is invariant to reparametrization. Empirically, we found that small region counts align with geometrically simple decision boundaries and correlate well with good generalization performance. We also observe that good hyper-parameter choices such as larger learning rates and smaller batch sizes can induce small region counts. We further establish the theoretical connections and explain how larger learning rate can induce small region counts in neural networks.

## 1 INTRODUCTION

One mystery in deep neural networks lies in their ability to generalize, despite having significantly more learnable parameters than the number of training examples (Zhang et al., 2017a). The choice of network architectures, including factors such as nonlinearity, depth, and width, along with training procedures like initialization, optimization algorithms, and loss functions, can result in vastly diverse generalization performances (Sutskever et al., 2013; Smith et al., 2017; Wilson et al., 2017; Li et al., 2019). The varied generalization abilities exhibited by neural networks are often explained by many researchers through the theory of *implicit bias* (Brutzkus et al., 2017; Soudry et al., 2018). Implicit bias refers to inherent tendencies of how the network learns and generalizes from the training data, even without explicit regularizations or constraints.

The implicit bias of linear neural networks has been extensively studied. One of the classical setting is linear classification with logistic loss. Brutzkus et al. (2017); Soudry et al. (2018); Arora et al. (2019) show that the parameter converges to the direction that maximizes the $L_2$ margin. For regression problems, it is proved that gradient descent or stochastic gradient descent converges to a parameter that is closest to the initialization in terms of $L_2$ norm (Gunasekar et al., 2018a). The results from the linear regression model can be extended to deep linear neural networks by generalizing the definition of min-norm and max-margin solutions (Ji & Telgarsky, 2018a; Vaskevicius et al., 2019; Woodworth et al., 2020).

Compared to linear models, defining implicit biases in non-linear networks poses significant challenges. One line of work studies homogeneous networks and demonstrates that gradient flow solutions converge to a KKT point of the max-margin problem (Lyu & Li, 2019; Ji & Telgarsky, 2020; Wang et al., 2021; Jacot et al., 2022). Further research extends this analysis, showing that gradient flow converges to a max-margin solution under various norms (Ongie et al., 2019; Chizat & Bach, 2020). Other studies focus on describing the implicit bias of neural networks using sharpness, such as (Foret et al., 2020; Montúfar et al., 2022; Andriushchenko et al., 2023).

We note that previous definitions of implicit bias in neural networks mostly focus on certain metrics of network parameters. Such approaches enable explicit analyses of training trajectories, but face new challenges when applied to nonlinear networks: reparametrization of the network may preserve

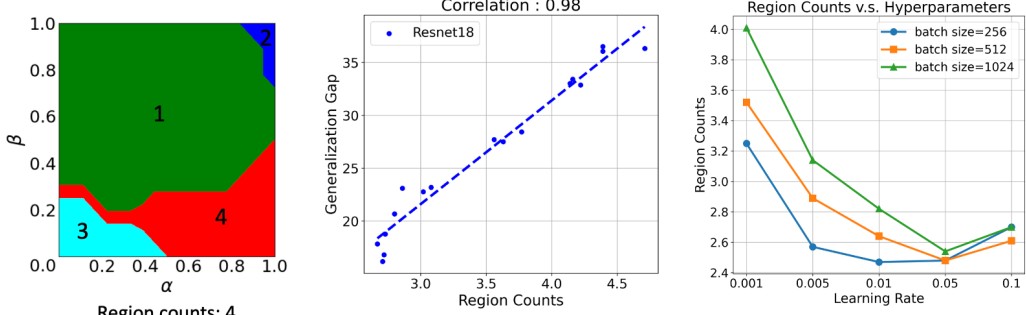

**Figure 1: A schematic illustration of main results in this paper.** Left: The region counts in 2-dimension input space. Each distinct region represents an area where the neural network makes the same prediction for all points within that region. Middle: A strong correlation between region counts and the generalization gap. Right: Larger learning rate or smaller batch size induces smaller region counts.

the function mapping but gives completely different parameters, and consequently, different implicit biases. We will discuss this point in detail in Section 3.

Motivated by the above studies, we instead focus on leveraging the decision boundaries in the input space to characterize implicit bias. Our research identifies a metric called region count, which is defined by the average number of regions in the predictor's decision boundary (See Figure 1). We select a low-dimensional space, use the neural network to predict the labels of all points within this space, and define the number of connected components with the same label as the region count in that subspace. This definition differs from previous studies on the number of linear regions (Hanin & Rolnick, 2019a;b; Safran et al., 2022) which is defined as the set of inputs that correspond to the same activation pattern in the network. We find that this region count has a strong correlation with the generalization gap, defined as the difference in percentage between the training and test errors. The experiments in Figure 1 suggests that models with fewer region counts tend to generalize better.

Furthermore, we show that neural networks trained with large learning rate or small batch size, which are typically deemed as beneficial for generalization, are biased towards solutions that have small region counts. Therefore, region count empirically serves as an accurate generalization metric as well as an implicit bias indicator. We also provide theoretical analyses to explain this phenomenon. We prove that for two-layer ReLU neural networks, gradient descent with large learning rate induces a small region count, which accords well with our empirical findings.

The main contributions of this paper are listed as follows:

1. We introduce a novel measure of implicit bias via the region count in the input space. Through extensive experiments, we verify a strong correlation between region count and the generalization gap. This correlation remains robust across different learning methods, datasets, training parameters, and counting methods.

2. We assess the factors that induce small region count, discovering that training with larger learning rates and smaller batch sizes typically results in fewer regions. This provides a possible cause for the implicit bias in neural networks.

3. We conduct theoretical analyses on region counts, and show that for two-layer ReLU neural networks, gradient descent with large learning rate induces a small region count.

## 2 RELATED WORKS

**Implicit Bias of Linear Neural Network**    The implicit bias in linear neural networks are thoroughly investigated in recent works. For linear logistic regression on linearly separable data, full-batch gradient descent converges in the direction of the maximum margin solution (Soudry et al., 2018). This foundational work has various follow-ups, including extensions to non-linearly-separable data (Ji & Telgarsky, 2018b; 2019), stochastic gradient descent (Nacson et al., 2019), and other loss functions and optimizers (Gunasekar et al., 2018a).

These findings in linear logistic regression are generalized to deep linear networks. For fully-connected neural networks with linear separable data, Ji & Telgarsky (2018a) show that the direction of weight also converges to $L_2$ max-margin solution. For linear diagonal networks, the gradient flow maximizes the margin with respect to a specific quasi-norm that is related to the depth of network (Gunasekar et al., 2018b; Woodworth et al., 2020; Pesme et al., 2021), leading to a bias towards sparse linear predictors as the depth goes to infinity. This sparsity bias also exists in linear convolutional networks (Gunasekar et al., 2018b; Yun et al., 2020).

**Implicit Bias of Non-linear Neural Network**   The non-linearity of modern non-linear neural networks pose challenges to studying its implicit bias. Initial works in this area (Lyu & Li, 2019; Ji & Telgarsky, 2020) focus on homogeneous networks. These studies show that with exponentially-tailed classification losses, both gradient flow and gradient descent converge directionally to a KKT point in a maximum-margin problem. Further studies, for instance in (Wang et al., 2021), consider a more general setup that includes different optimizers and prove that both Adam and RMSProp are capable of maximizing the margin in neural networks while Adagrad is not. Ongie et al. (2019); Chizat & Bach (2020) showcased a bias towards maximizing the margin in a variation norm for infinite-width two-layer homogeneous networks. Lyu et al. (2021); Jacot et al. (2022) identified margin maximization in two-layer Leaky-ReLU networks trained with linearly separable and symmetric data. More recent investigations into non-linear neural networks, such as (Jacot, 2022), focus on the homogeneity of the non-linear layer, demonstrating an implicit bias characterized by a novel non-linear rank.

**Region Counts of Neural Network**   Many previous works focus on calculating the linear regions of neural networks (Hanin & Rolnick, 2019a;b). A linear region is a set of inputs that share the same activation pattern in the network. Safran et al. (2022) proves that for a two-layer ReLU network with width $r$, gradient flow will converge directionally to a network characterized by no more than $O(r)$ linear regions. Serra et al. (2018) and Cai et al. (2023) explain that maximizing the number of linear regions can lead to better-performing networks, and they explore how network structures can be designed to achieve more linear regions. The number of linear regions is independent of the network's label output, focusing more on its representational capacity rather than generalization ability. In contrast, we define decision regions as connected areas in the input space that correspond to the same label, and our work explores the relationship between the number of regions and the generalization gap.

The paper (Nguyen et al., 2018) is the most relevant to our work, as it similarly defines decision regions corresponding to label predictions of neural networks. They define decision regions in the entire input space, whereas our definition and counting method focus on a subspace. Their conclusion states that, under certain conditions, each label has only one connected decision region in the entire space. However, two points being connected in the entire space does not imply they are connected in a subspace. For instance, two points may be connected in a three-dimensional space, but a two-dimensional cross-section may not provide a connecting path. Thus, we observe multiple regions in 1D or 2D subspaces and find a strong correlation with generalization ability.

## 3 MOTIVATION

Norm-based and margin-based characterizations belong to the most popular measures of implicit bias. Various definitions for norm and margin exist. For simplicity, we consider the following two definitions.

**Example 1** (Norm and Margin). *Let $W = \{W_1, \cdots, W_l\}$ denote the post-training weight parameters of an $l$-layer neural network $f_W(x) = W_l\sigma(W_{l-1}\cdots W_2\sigma(W_1x))$, with $\sigma(\cdot)$ as the ReLU activation function. Denote the weight initialization as $W^0 = \{W_1^0, \cdots, W_l^0\}$. Consider the Frobenious norm between network weights and initialization:*

$$d(W) = \sqrt{\sum_{i=1}^{l} \|W_i - W_i^0\|_F^2},$$

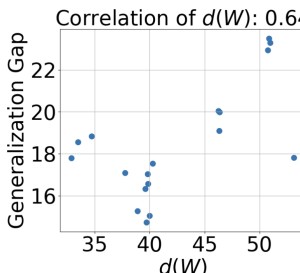 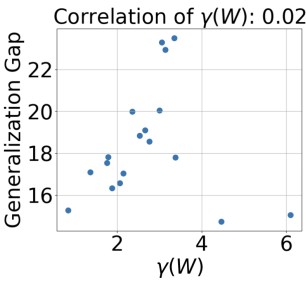 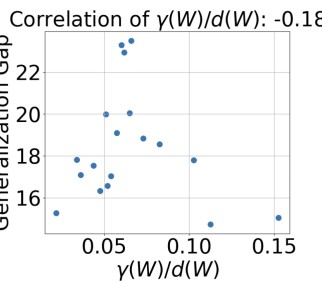

**Figure 2: Norm-based and margin-based measures may not be predictive of generalization gaps.** We train ResNet18 on the CIFAR-10 dataset using various hyperparameters. These implicit bias measures can be ineffective for general non-linear neural networks.

*and the output-space margin*

$$\gamma(W) = \mathbb{E}_{(x,y) \in D_{train}} \left[ f(x)_y - \max_{i \neq y} f(x)_i \right].$$

$d(\cdot)$ and $\gamma(\cdot)$ are commonly used indicator for implicit bias of linear models (Soudry et al., 2018; Ji & Telgarsky, 2018b). However, both of them are not invariant to network reparameterization. We can construct a different set of network weight parameters by scaling the parameters as $\hat{W} = \{2W_1, \frac{1}{2}W_2, \cdots, W_l\}$, such that $f_W = f_{\hat{W}}$ but $d(W) \neq d(\hat{W})$ in general. Similarly, we can scale the last layer weights and get $\tilde{W} = \{W_1, \cdots, 2W_l\}$, such that $\gamma(\tilde{W}) \neq \gamma(W)$, but $\text{argmax}_i f_W(x) = \text{argmax}_i f_{\tilde{W}}(x)$. This reparameterization trick also works for more complicated norm-based and margin-based generalization metric in (Jiang et al., 2019), or the sharpness-based metrics (Andriushchenko et al., 2023).

We numerically investigate whether they are effective measures, by training a ResNet18 on Cifar10 dataset, using different hyperparameters as in Table 1. The results are presented in Figure 2, which indicates that these measures have a low correlation with the generalization gap in the deep learning regime. One could choose other definitions of norms to achieve stronger correlations, but such choices are often problem-specific and require domain expertise, as discussed in (Jiang et al., 2019).

The definition of margin may also be improved to the input-space margin, *i.e., the $\ell_2$ distance of input data $x$ to decision boundary defined by the classifier,* which is able to characterize the quality and robustness of the classifier. This metric is invariant to reparameterization and therefore more intrinsic to the underlying classifier. However, due to the highly nonconvex loss landscape, the input-space margin is NP-complete to compute and even hard to approximate (Katz et al., 2017; Weng et al., 2018). Therefore, quantitatively analyzing the decision boundary of a neural network and characterizing its implicit bias remains a challenge.

Our motivation can be summarized by a simple idea: although the margin in the input space is hard to compute, we can quantify the regions split by the decision boundary. This measure is invariant to model reparametrization and can also capture the complexity of the decision boundary. This motivates us to consider the region counts as an implicit bias metric.

## 4 PRELIMINARY

Although region count is a natural measure for the complexity of a predictor, and it depends only on the decision function rather than the model parameterization, its formal definition and computability remains unclear. In this section, we first provide the definition and low-dimensional approximation of region counts. We then empirically verify that region counts correlate with generalization gap.

### 4.1 DEFINITION OF REGION COUNTS

Let $d$ denote the training data dimension and $f : \mathbb{R}^d \rightarrow \{1, 2, \ldots, N\}$ denote a neural network for a classification task with $N$ classes. For a subset $U \subset \mathbb{R}^d$, we can define the connectedness of its element as follows:

**Definition 1** (Connectedness). We say the data points $x_1, x_2 \in U$ are (path) connected with respect to a neural network $f$ if they satisfy:

- $f(x_1) = f(x_2) = c$,

- There exist a continuous mapping $\gamma : [0, 1] \rightarrow U$, $\gamma(0) = x_1$, $\gamma(1) = x_2$, and for any $t \in [0, 1]$, $f(\gamma(t)) = c$.

Then we define the connected region in this subset:

**Definition 2** (Maximally Connected Region). We say $V \subset U$ is a maximally connected region in $U \in \mathbb{R}^d$ with respect to a neural network $f$ if it satisfies the following property:

- For any $x, y \in V$, they are connected.

- For any $x \in V$, $y \in U \setminus V$, they are not connected.

Finally, we formally define the region count as follows:

**Definition 3** (Region Count). For a subset $U \subseteq \mathbb{R}^d$, we define its region count $R_U$ as the number of maximally connected regions in $U$ with respect to a neural network $f$:

$$R_U = \text{card}\{V \subset U | V \text{ is a maximally connected region}\},$$

where card is the cardinality of a set.

### 4.2 ESTIMATING REGION COUNTS

Calculating the region count in the original high-dimensional input space can be computationally intractable. Therefore, we propose a computationally efficient surrogate by calculating the region counts on low dimensional subspace spanned by training data points.

**Definition 4** (Region Count in $d$-Dimensional Subspace). We randomly sample $d + 1$ datapoints in the training set $D_{train}$ to generate a convex region in $\mathbb{R}^d$ subspace. The $d$-dimensional region count $R_d$ is defined as the expectation of number of maximally connected regions:

$$R_d = E_{x_1, x_2, \ldots, x_{d+1} \sim D_{train}}[R_{\text{Conv}\{x_1, x_2, \ldots, x_{d+1}\}}],$$

where $x_1, x_2, \ldots, x_{d+1}$ are sampled from the training dataset, and $\text{Conv}\{x_1, x_2, \ldots, x_{d+1}\}$ is the convex hull formed by these $d + 1$ points.

This paper primarily focuses on low dimension spaces, which is illustrated as below. In practice, we randomly sample training data points for multiple times and take the average region counts. In Section 7, we show that the choice of subspace dimension $d$ does not significantly affect the results. The details on how to count the regions and generate the polytopes are provided in Appendix B.

**Example 2** (Region counts in 1D and 2D subspace). *For region count in 1-dimensional subspace, we randomly sample two data points, denoted as $x_1$ and $x_2$, from the training set, and calculate the region count on the line segment connecting them:*

$$\{\alpha x_1 + (1 - \alpha)x_2, 0 \le \alpha \le 1\}.$$

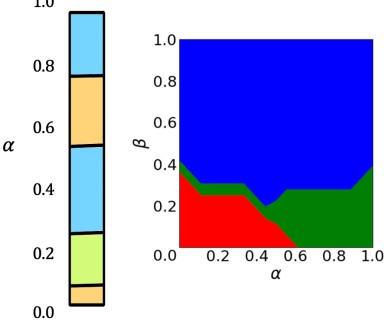

1D Region Counts: 5    2D Region Counts: 3

**Figure 3: Illustrations of region counts in 1D and 2D subspace.** We use different colors to represent different outputs of the neural network.

*For the 2-dimensional case, we randomly sample three data points, $x_1$, $x_2$, and $x_3$, from the training set, and calculate the region count in the convex hull spanned by them:*

$$\{\alpha x_1 + \beta x_2 + (1 - \alpha - \beta)x_3, \alpha \ge 0, \beta \ge 0, \alpha + \beta \le 1\}.$$

*We provide an illustration in Figure 3.*

## 5 REGION COUNTS CORRELATE WITH GENERALIZATION GAPS

In this section, we present our major empirical findings, which reveal a strong correlation between region counts and the generalization error of neural networks.

We conduct image classification experiments on the CIFAR-10 dataset, using different architectures, including ResNet18 (He et al., 2016), EfficientNetB0 (Tan & Le, 2019), and SeNet18 (Hu et al., 2018). Results on other architectures are deferred to ablation studies. We vary the hyperparameters for training, such as learning rate, batch size and weight decay coefficient, whose numbers are reported in Table 1. The region count is calculated using randomly generated 1D hyperplanes, as described in Example 2. We run each experiment 100 times and report the average number.

We plot the region count and generalization gap of different setups in Figure 4, and calculate the correlation between them. For each network architecture, we observe a strong correlation as high as 0.98. The overall correlation for all the three networks still reaches 0.93. This reveals a remarkably high correlation between region counts and generalization gap.

We also explore whether such a strong correlation exists for traditional machine learning algorithms. We conduct experiments with decision trees and random forests on the same classification tasks, with hyperparameters specified in Table 1. We observe a similar linear trend between region count and generalization gap, with a correlation of 0.96 in decision trees and 0.98 in random forests. The overall correlation coefficient is 0.90. Therefore, region count serves as a good indicator for generalization performance across various setups.

**Table 1: The hyperparameters for experiments.** Left: We vary the learning rate, batch size, and weight decay for training a neural network, to modulate the model's generalization ability. Right: We adjust the training parameters for traditional machine learning models, such as decision tree and random forest.

| Hyperparameters | Value | Hyperparameters | Value |
|---|---|---|---|
| Learning rate | $0.1, 0.01, 0.001$ | Depth | $3, 4 \cdots, 17$ |
| Batch size | $256, 512, 1024$ | Criterions | gini, entropy |
| Weight decay | $10^{-5}, 10^{-6}, 10^{-7}$ | Splitter | best, random |

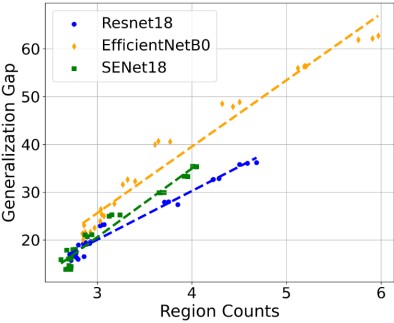 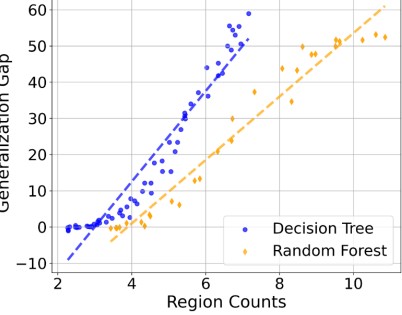

**Figure 4: Strong correlation between region counts and generalization gap.** Left: We conduct experiments using three neural networks on the CIFAR-10 dataset, with various hyperparameters. There is a strong correlation between region counts and the generalization gap, with a correlation coefficient of 0.98 for each network and 0.93 across all networks. Right: We conduct experiments using Decision Tree and Random Forest on the CIFAR-10 dataset. The result also reveals a strong correlation between region counts and the generalization gap.

## 6 REGION COUNTS QUANTIFY IMPLICIT BIAS

In this section, we further investigate the implicit bias of neural networks via region counts. We show both empirically and theoretically that neural networks trained with appropriate hyperparameters tend to have smaller region counts, thus achieving better generalization performance.

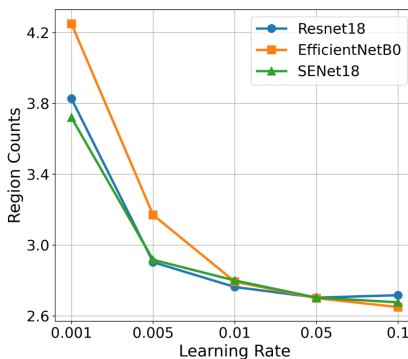 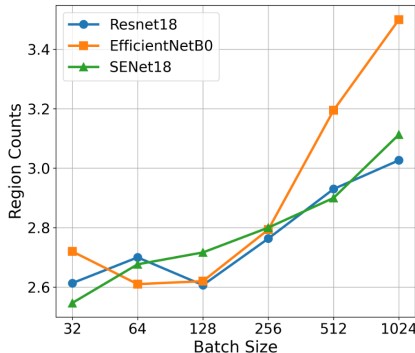

**Figure 5: Large learning rate and small batch size reduce region counts.** We train three networks on the CIFAR-10 dataset, varying the batch sizes and learning rates. Our findings reveal that a smaller batch size or a higher learning rate results in smaller region counts, allowing the network to learn a simpler decision boundary and generalize better.

### 6.1 THE BIAS FROM TRAINING HYPERPARAMETERS

Training neural networks requires careful selection of many hyperparameters, such as learning rates, batch sizes, optimizers, epochs and so on. Here, we primarily focus on learning rate and batch size, and study their impact on the region count.

**Learning Rates.** We provide the relationship between the learning rate and the region count in Figure 5. Our findings indicate that a larger learning rate tends to simplify the decision boundary and results in a smaller region count in the hyperplane. This accords well with real practices, where large learning rates of 0.1 or 0.01 are often favored for better generalization.

**Batch Sizes.** Similarly, the training batch size can impact the number of regions. As shown in Figure 5, smaller batch sizes lead to a model with fewer regions. This result reveals the advantage of small-batch training, which leads to better generalization accuracy.

Previous studies (Keskar et al., 2016; Jastrzebski et al., 2017; Hoffer et al., 2017; Novak et al., 2018) find that certain hyperparameters, such as a large learning rate and a small batch size, can improve the generalization of the neural network. Our observations provide a possible explanation: these good hyperparameter choices lead to a reduced region count. Such simplicity bias can decrease the generalization gap of neural networks.

### 6.2 THEORETICAL EXPLANATIONS

Next, we present a theoretical analysis to explain why some hyperparameter choices, such as large learning rate, can lead to small region counts.

Consider a two layer ReLU neural network $f_W(x) = \sum_{i=1}^p a_i \sigma(w_i^\top x)$. The second layer weights $a_i$ are initialized uniformly from $\{1, -1\}$ and fixed throughout training. Let $\mathcal{D} = \{(x_i, y_i)\}_{1 \le i \le N}$ denote the training set. Consider training $f_W$ on $\mathcal{D}$ using gradient descent (GD) with learning rate $\eta$. We choose the quadratic loss $l(W, x, y) = \frac{1}{2}(y - f_W(x))^2$ and denote $L(W) = \frac{1}{N} \sum_{i=1}^N l(W, x_i, y_i)$. Denote the GD trajectory as $\{W_i\}_{i \ge 0}$. For two input data $x_a, x_b$, Let $R(x_a, x_b, W)$ denote the region count on the line segment connecting them, and $N(x_a, W)$ denote the number of activated neurons with input $x_a$, i.e., the number of $i$ such that $w_i^\top x_a > 0$.

We make the following assumption on the data distribution.

**Assumption 1.** The training dataset $\mathcal{D} = \{(x_i, y_i)\}_{1 \le i \le N}$ satisfies the following two properties:

1. $\|x_i\| \ge r$ for all $1 \le i \le N$,

2. With probability one, any $W \in \{W_i\}_{i \ge 0}$ satisfies $w_i^\top x_j \ne 0$ for all $1 \le i \le q, 1 \le j \le N$, where the randomness comes from weight initialization.

The validity of Assumption 1 comes from the fact that the bifurcation zone (Bertoin et al., 2021) of ReLU neural networks, which contains its non-differentiable points, has Lebesgue measure zero (Bolte & Pauwels, 2020; 2021; Bianchi et al., 2022). Therefore, if the distribution of weights are absolutely continuous with respect to the Lebesgue measure, the bifurcation zone can be avoided with probability one. We conjecture that it can be proved rigorously, but leave it as an assumption since the proof diverges from the main content in this paper.

The next assumption characterizes the sharpness along the training trajectory. This is actually from the well-known edge of stability phenomenon (Cohen et al., 2020; Damian et al., 2022; Arora et al., 2022; Ahn et al., 2024), which states that the sharpness of neural networks, characterized by the $\ell_2$ norm of the Hessian matrix, hovers around $\frac{2}{\eta}$.

**Assumption 2** (Edge of Stability). *There exist a $T \in \mathbb{N}$, such that for $t \geq T$, with we have*

$$\lambda_{\max}(\nabla_W^2 L(W_t)) = \Theta\left(\frac{1}{\eta}\right),$$

*where $\lambda_{\max}$ denotes the maximum eigenvalue of a matrix.*

We are now ready to present the main theorem, which establishes a relationship between region count and learning rate.

**Theorem 1.** *Under Assumption 1 and 2, we have that for neural net weights $W_t$ at training step $t \geq T$ with probability one, the average region count $R(X, X', W_t)$ for random training data point $X, X'$ can be bounded as:*

$$\mathbb{E}_{X,X'}[R(X, X', W_t)] = \frac{1}{N^2} \sum_{i=1}^{N} \sum_{j=1}^{N} R(x_i, x_j, W_t) \leq O\left(\frac{N}{r^2\eta}\right).$$

The theorem demonstrates that with a larger learning rate, gradient descent has the implicit bias to yield solutions with smaller region counts. This aligns well with the previous observations.

We defer the proof of this theorem to Appendix D, and sketch the proof as follows. The proof begins with bounding the region count using the activation pattern of ReLU neurons, as stated in the following lemma.

**Lemma 2.** *The region counts between a pair of data points is upper-bounded by the number of active neurons. For two inputs $x_a, x_b$, we have $R(x_a, x_b, W) \leq N(x_a, W) + N(x_b, W) + 2$.*

Then we prove that the activation pattern gives a bound on the smoothness of the training loss.

**Lemma 3.** *The sharpness of a neural network is lower-bounded by the number of active neurons:* $\lambda_{\max}\left(\nabla_W^2 L(W)\right) \geq \frac{r^2}{N^2} \sum_{i=1}^{N} N(x_i, W)$.

Note that this lemma brings in an additional $N$ in the denominator, which leads to a $N$-dependent bound in Theorem 1. We conjecture that the $N$-dependency can be optimized under further structural assumptions on the data distribution, and leave it for further investigations. Equipped with these two lemmas, Theorem 1 is a consequence of the sharpness condition in Assumption 2.

## 7 ABLATION STUDIES

This section presents an ablation study to validate the robustness and consistency of our findings. We systematically vary key aspects of our experimental setup, including the network architecture, dataset, optimizer, and the method of computing the plane, and evaluate their impact on our main results of the correlation between region count and the generalization gap.

**More Architectures, Datasets and Hyperplane Dimensions.** We first examine the influence of neural network architectures and datasets on our results. We provide additional results on various neural network architectures such as ResNet34 (He et al., 2016), VGG19 (Simonyan & Zisserman, 2014), MobileNetV2 (Sandler et al., 2018), ShuffleNetV2 (Ma et al., 2018), RegNet200MF (Radosavovic et al., 2020), and SimpleDLA (Yu et al., 2018). We also use various datasets such as

CIFAR-100 (Krizhevsky et al., 2009) and ImageNet (Deng et al., 2009). Region counts and generalization gaps are evaluated across various learning rates, batch sizes, and weight decay parameters as listed in Table 1.

We also explore the effects of different methods for generating the hyperplane in the input space. In our previous experiments, we generate the 1-dimensional plane using random pairs of samples from the training set and calculate the region count on them. Here we explore region counts in higher dimensional planes, that are spanned by 2 to 5 data randomly-selected points from the training set, using the CIFAR-10 dataset.

The experiment results of the correlation are presented in Table 2. We also provide correlation plots for each network in Appendix A.2. We observe that the strong correlation between region count and the generalization gap remains consistent in various setups. The consistency indicates that our findings reveal a fundamental characteristic of non-linear neural networks.

We also provide evaluations by varying the optimizer and hyperplane generation algorithms. The results are deferred to Appendix C.

**Table 2: Experimental consistency across networks, datasets, and counting methods.** We conduct experiments on various types of networks across multiple datasets. We also alter the method of calculating the region counts. The results of the correlation indicate that our findings are consistent across different setups.

| Network | Dataset | | | Counting Dimension | | | |
|---|---|---|---|---|---|---|---|
| | *CIFAR-10* | *CIFAR-100* | *ImageNet* | 2 | 3 | 4 | 5 |
| ResNet18 | 0.98 | 0.96 | 0.91 | 0.96 | 0.97 | 0.97 | 0.96 |
| ResNet34 | 0.98 | 0.98 | 0.82 | 0.98 | 0.98 | 0.98 | 0.99 |
| VGG19 | 0.94 | 0.85 | 0.78 | 0.88 | 0.86 | 0.84 | 0.86 |
| MobileNet | 0.95 | 0.95 | 0.92 | 0.99 | 0.99 | 0.99 | 0.99 |
| SENet18 | 0.98 | 0.85 | 0.80 | 0.97 | 0.97 | 0.97 | 0.93 |
| ShuffleNetV2 | 0.95 | 0.92 | 0.92 | 0.94 | 0.95 | 0.95 | 0.93 |
| EfficientNetB0 | 0.98 | 0.84 | 0.93 | 0.99 | 0.99 | 0.99 | 0.98 |
| RegNetX_200MF | 0.98 | 0.87 | 0.97 | 0.98 | 0.99 | 0.99 | 0.98 |
| SimpleDLA | 0.98 | 0.94 | 0.84 | 0.99 | 0.99 | 0.98 | 0.99 |

**Data Augmentations.** Mixup (Zhang et al., 2017b) is a data augmentation technique that creates training samples by linearly interpolating pairs of input data and their corresponding labels. We train a ResNet-18 model using mixup, with other hyperparameters in Table 1. The plot in Figure 6 illustrates that Mixup induces smoother decision boundaries with smaller region count and has a better generalization performance.

Random crop and random horizontal flip is another way to enhance the diversity of the training dataset. We apply random crop of size $32\times32$ with padding 4 and random horizontal flip with a probability of 0.5 as data augmentations. As depicted in Figure 7, we observe that compared with mixup, random crop and random flip result in a more evident vertical shift in the performance curve.

Both Figure 6 and Figure 7 show that employing these techniques does not alter the correlation between region counts and generalization gaps.

**Evolution of Region Counts during Training.** Next we study how the region count, generalization gap and their correlation evolve during training. Following the setup in Section 5, we train a ResNet18 model on CIFAR-10 dataset, and report the region count and generalization gap during the training process. The statistics are averaged over different hyperparameter choices as in Table 1.

The results are provided in Table 3. We recorded the average values of region count and generalization gap for these data points in the second and third columns to show their changes during the training process. We observe that the correlation is very low at initialization, but steadily increases during training. This suggests that the metric of region count is not a property of the neural network initialization, but rather inherently involved with the neural network training algorithm.

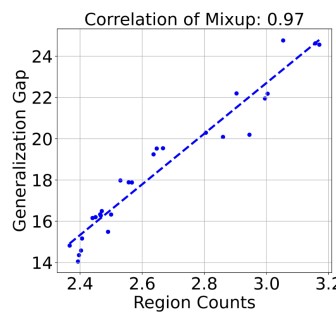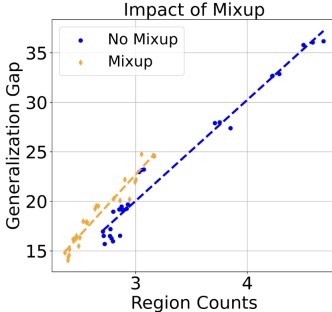

**Figure 6: The impact of mixup.** This figure shows that mixup improves the model's generalization ability and reduces the number of regions in the hyperplane.

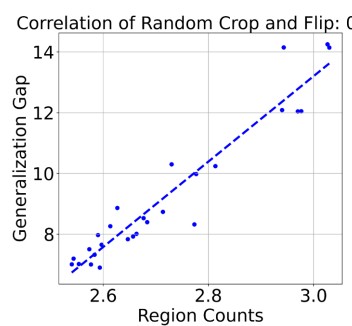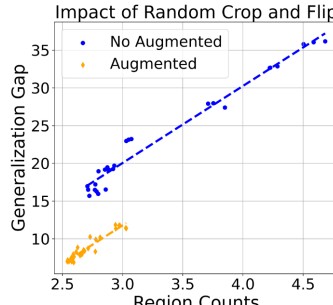

**Figure 7: The impact of random crop and random flip.** Unlike mixup, data augmentation results in a vertical shift in the performance curve, accompanied by a decrease in the number of regions and a more significant enhancement in test accuracy.

**Table 3: The evolution of region count, generalization gap and their correlation.** The correlation is very low at initialization, but steadily increases during training.

| Training Epoch | Region Counts | Generalization Gap | Correlation |
|---|---|---|---|
| 0 | 1.13 | N/A | N/A |
| 20 | 3.14 | 11.2 | -0.53 |
| 40 | 3.02 | 7.3 | -0.29 |
| 60 | 3.10 | 18.2 | 0.35 |
| 80 | 3.22 | 26.7 | 0.77 |
| 100 | 3.25 | 30.4 | 0.98 |
| 130 | 3.28 | 31.2 | 0.97 |
| 160 | 3.27 | 31.7 | 0.98 |
| 200 | 3.26 | 31.6 | 0.98 |

## 8 CONCLUSIONS AND FUTURE DIRECTIONS

This paper introduces a novel approach to characterizing the implicit bias of neural networks. We study the region counts in the input space and identify its strong correlation with generalization gap in non-linear neural networks. These findings are consistent across various network architectures, datasets, optimizers. Our analysis offers a new perspective to quantify and understand the generalization property and implicit bias of neural networks.

Our paper suggests several promising directions for future research. Firstly, our analyses of why large learning rate induces small region counts mainly focus on a simplified setup. The analyses for more general settings remain open. Secondly, extending the definition of region count to non-classification tasks, such as natural language generation, would be a worthwhile direction. Lastly, region count can be leveraged to design new architectures or regularization, that can potentially improve the generalization performance of neural networks.

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

## A    EXPERIMENT DETAILS

In this section, we provide the detailed experiment settings.

### A.1    DETAILS ON ARCHITECTURES AND DATASETS

We conduct experiments on different neural network architectures, including ResNet18 and ResNet34 (He et al., 2016), EfficientNetB0 (Tan & Le, 2019), SENet18 (Hu et al., 2018), VGG19 (Simonyan & Zisserman, 2014), MobileNetV2 (Sandler et al., 2018), ShuffleNetV2 (Ma et al., 2018), RegNet200MF (Radosavovic et al., 2020), SimpleDLA (Yu et al., 2018). We conduct all experiments using NVIDIA RTX 6000 graphics card.

We use CIFAR-10/100 (Krizhevsky et al., 2009) and Imagenet-1k (Deng et al., 2009) as datasets. For CIFAR-10 and CIFAR-100 dataset, each network was trained for 200 epochs using the Stochastic Gradient Descent (SGD) algorithm with cosine learning rate schedule. We choose 27 combinations of hyperparameters in Table 1, and for each hyperparameter we use 3 random seeds and report the average metrics. For the Imagenet-1k dataset, each network was trained for 50 epochs with random data crop and random flip. We use the same optimizer and 27 combinations of hyperparameters as in CIFAR-10 and CIFAR-100 experiments. It is worth noting that we make minor adjustments on hyperparameters for certain networks to ensure stable training. For example, in the case of VGG19, the training is unable to converge when the learning rate is set to 0.1; therefore, we adjust it to 0.05.

### A.2    CORRELATION PLOTS

We show the correlation plot of average regions and test accuracy in Figure 8. The figure consists of results from training different networks on CIFAR-10 dataset with SGD for 200 epochs, using the hyperparameters specified in Table 1. The results show that different networks have different number of regions, ranging from 2 to 20. However, the correlation of test accuracy and average number of regions are consistently high in all the networks.

## B    HOW TO CALCULATE THE NUMBER OF REGIONS

In this section, we study different methods to calculate the region count, and discuss their impact on the results. Since it is impossible in practice to calculate the predictions of an infinite number of data points on the hyperplane, we select grid points from the hyperplane to calculate the region count.

Assume we have divided a region of the hyperplane into several equidistant small squares. We can use an algorithm similar to breadth-first search to calculate the number of connected components within these small squares, thereby determining the number of regions. Here, we use a 2-dimensional hyperplane as an example (the 1-dimensional case can be considered a degenerate version of this algorithm). The algorithm for calculating the number of regions in this setup is given in Algorithm 1.

Therefore, it is necessary to determine the granularity of splits for the plane. We experimented with different setups of splitting parameters, and the results averaged by 100 independent trials are presented in Table 4. From the results, using 200 grid points in the 1D case and 30x30 grid points in the 2D case is an optimal choice. Splitting the plane into fewer points results in an inadequate approximation of regions, while increasing the number of points does not significantly enhance accuracy but incurs greater computational costs. Therefore, in our experiments, we split the plane into 200 grid points for the 1D case and 30x30 grid points for the 2D case.

Subsequently, we study the number of random samples in calculating the average number of regions. We experiment with different numbers of hyperplanes, and the results are presented in Table 5. From the results, we know that using 100 samples to calculate the average provides a reliable answer with relatively low computational costs. Therefore, in the experiments we randomly generate 100 lines or planes and calculate the average number of regions.

In our paper we use the convex hull of two points $\{\alpha x_1 + (1 - \alpha x_2)\}$ to calculate the region counts in 1D case with $\alpha \in [0, 1]$. We also conduct ablation studies with varied coordinate ranges $\alpha$. We train ResNet18 on CIFAR10 using hyperparameter in Table 1 in our manuscript, where we vary the range of $\alpha$ and analyze the correlation. The results are shown in Table 6. These studies confirm

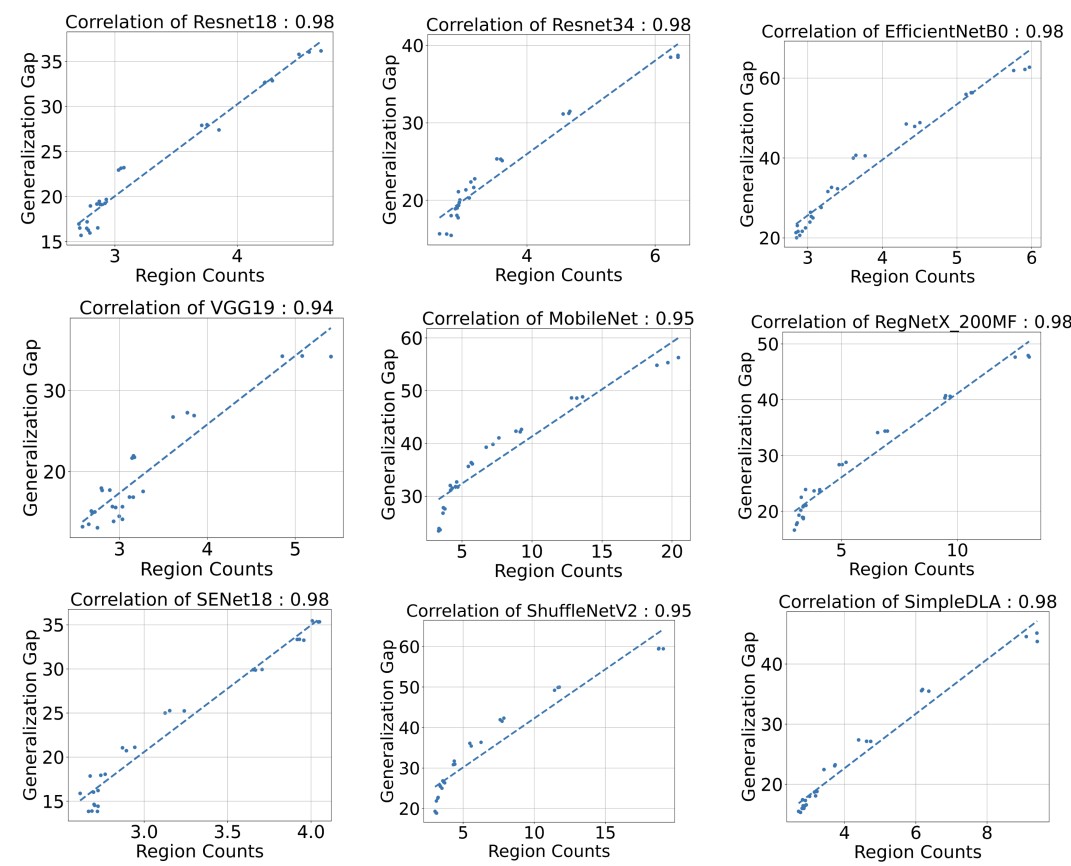

**Figure 8:** The correlation plot of all networks between average regions and test accuracy for CIFAR-10 dataset with optimizer SGD. From the graph we know that the correlations are all very high for different non-linear networks. Different structures of neural networks incur different scope of the average regions.

---

**Algorithm 1** Calculate the Number of Region

---

**Input:** Prediction Matrix $P$ with dimension$(w, h)$.
**Output:** Number of connected regions $N$.
 1: Initialize a mark matrix $M$ to a zero matrix, with the same dimensions as $P$
 2: Initialize count of connected regions $N \leftarrow 0$
 3: **for** $i \leftarrow 0$ **to** $w - 1$ **do**
 4:   **for** $j \leftarrow 0$ **to** $h - 1$ **do**
 5:     **if** $M[i][j]$ is already marked **then**
 6:       **continue**
 7:     **end if**
 8:     Mark position $(i, j)$ in $M$ as visited
 9:     Perform Breadth-First Search (BFS) starting from position $(i, j)$
10:     In BFS, enqueue all neighboring points that have the same value as $(i, j)$ in $P$ and mark them as visited in $M$
11:     Continue BFS until the queue is empty
12:     Increment count of connected regions $N \leftarrow N + 1$
13:   **end for**
14: **end for**
15: **return** $N$

---

that expanding the range does not influence the strong correlation between region counts and test accuracy.

**Table 4:** The mean value of region counts with different splitting numbers in 1d (left) and 2d (right) planes.

| Splitting Numbers | Region Counts | | Splitting Numbers | Region Counts |
|---|---|---|---|---|
| 50 | 2.74 | | 10×10 | 2.76 |
| 100 | 2.76 | | 20×20 | 2.76 |
| 200 | 2.78 | | 30×30 | 2.78 |
| 300 | 2.78 | | 40×40 | 2.78 |
| 500 | 2.78 | | 50×50 | 2.78 |

**Table 5:** The mean value of region counts with different number of random samples.

| Number of Samples | Region Counts |
|---|---|
| 10 | 2.24 |
| 50 | 2.56 |
| 100 | 2.78 |
| 300 | 2.80 |
| 500 | 2.79 |

**Table 6:** The impact of interpolation range on region counts.

| The range of $\alpha$ | Region Counts | Correlation |
|---|---|---|
| $[0, 1]$ | 3.56 | 0.98 |
| $[-1, 2]$ | 4.47 | 0.96 |
| $[-2, 3]$ | 5.86 | 0.92 |
| $[-3, 4]$ | 6.39 | 0.93 |

## C  MORE ABLATION STUDIES

**Gradient Optimizers.**  We calculate the region count of models trained by different optimizers, including SGD, Adam, and Adagrad. The correlation between region count and the generalization gap is consistent for them, as detailed in Table 7.

**Table 7:** The impact of optimizers on the correlation between region counts and generalization gap.

| Optimizer / Network | SGD | Adam | Adagrad |
|---|---|---|---|
| ResNet18 | 0.98 | 0.92 | 0.96 |
| ResNet34 | 0.98 | 0.92 | 0.91 |
| VGG19 | 0.94 | 0.92 | 0.87 |
| MobileNet | 0.95 | 0.95 | 0.99 |
| SENet18 | 0.98 | 0.78 | 0.91 |
| ShuffleNetV2 | 0.95 | 0.83 | 0.99 |
| EfficientNetB0 | 0.98 | 0.97 | 0.99 |
| RegNetX_200MF | 0.98 | 0.95 | 0.99 |
| SimpleDLA | 0.98 | 0.95 | 0.88 |

**Hyperplane Generation Methods.**  We explore the effects of different methods for generating the hyperplane in the input space. In the main experiments, we generate a 1-dimensional plane using random pairs of samples from the training set and calculated the number of distinct regions between them. In this section, we apply various techniques for plane generation: selecting two data points from the test set, choosing one data point from the training set and extending it in a random direction by a fixed length. We calculate the number of regions for each of these setups. The results in Table 8 are consistent across different hyperplane computational approaches.

**Table 8:** The impact of calculation methods on the correlation between region counts and generalization gap.

| Counting / Network | Test | Train | Random |
|---|---|---|---|
| ResNet18 | 0.98 | 0.98 | 0.98 |
| ResNet34 | 0.98 | 0.96 | 0.94 |
| VGG19 | 0.94 | 0.89 | 0.78 |
| MobileNet | 0.95 | 0.94 | 0.88 |
| SENet18 | 0.98 | 0.96 | 0.99 |
| ShuffleNetV2 | 0.95 | 0.95 | 0.92 |
| EfficientNetB0 | 0.98 | 0.98 | 0.92 |
| RegNetX_200MF | 0.98 | 0.97 | 0.92 |
| SimpleDLA | 0.98 | 0.97 | 0.96 |

# D    PROOF

This section contains the proof of the theorem in this paper.

We first prove two lemmas.

**Lemma 2.** *The region counts between a pair of data points is upper-bounded by the number of active neurons. For two inputs $x_a, x_b$, we have $R(x_a, x_b, W) \leq N(x_a, W) + N(x_b, W) + 2$.*

*Proof.* If $R(x_a, x_b, W) \leq 2$, then the equation naturally holds. Next we consider $R(x_a, x_b, W) > 2$. From the definition of region count, one can find $R := R(x_a, x_b, W)$ points on the line segment between $x_a$ and $x_b$, such that the neural network gives different predictions. Denote these points as $\tilde{x}_1, \cdots, \tilde{x}_R$. We have

$$f_W(\tilde{x}_i) f_W(\tilde{x}_{i+1}) < 0, 0 \leq i \leq R - 1.$$

Consider $\tilde{x}_i, \tilde{x}_{i+1}, \tilde{x}_{i+2}$. Since the neural network gives alternating predictions on these three points, it is nonlinear and has activation sign changes on the line segment connecting them. Therefore, we can find a $1 \leq n(i) \leq p$, such that $(w_{n(i)}^\top \tilde{x}_i)(w_{n(i)}^\top \tilde{x}_{i+2}) < 0$.

We prove it by contradiction. IF for all $1 \leq n(i) \leq p$, such that $(w_{n(i)}^\top \tilde{x}_i)(w_{n(i)}^\top \tilde{x}_{i+2}) \geq 0$. Suppose $\tilde{x}_{i+1} = \lambda \tilde{x}_i + (1 - \lambda)\tilde{x}_{i+2}$, then we have

$$f_W(\tilde{x}_{i+1}) = \sum_{i=1}^p a_i \sigma(w_i^\top x_{i+1}) = \sum_{i=1}^p a_i \sigma(w_i^\top x_{i+1})$$

$$= \sum_{i=1}^p a_i [\lambda \sigma(w_i^\top x_i) + (1 - \lambda)\sigma(w_i^\top x_{i+2})]$$

$$= \lambda f_W(\tilde{x}_i) + (1 - \lambda)f_W(\tilde{x}_{i+2}).$$

Therefore $f_W(\tilde{x}_{i+1})$ has the same sign of $f_W(\tilde{x}_i)$ and $f_W(\tilde{x}_{i+2})$, contradict with the condition that they have alternative signs. So we can find a $1 \leq n(i) \leq p$, such that $(w_{n(i)}^\top \tilde{x}_i)(w_{n(i)}^\top \tilde{x}_{i+2}) < 0$.

Since $w_{n(i)}^\top x$ is linear in $x$, this implies that

$$(w_{n(i)}^\top x_a)(w_{n(i)}^\top x_b) < 0.$$

We also prove it by contradiction. If they have the same sign then the convex combination of them have the same sign so $(w_{n(i)}^\top \tilde{x}_i)(w_{n(i)}^\top \tilde{x}_{i+2}) \geq 0$.

We have the following two observations about $n(i)$. Firstly, we can choose an $n(i)$ such that $a_{n(i)} w_{n(i)}^\top \tilde{x}_{i+2}$ and $f_W(\tilde{x}_{i+2})$ have the same sign, since there exists at least one such neuron that contributes to the sign change of $f_W$. This implies that $n(i) \neq n(i+1)$, since $f_W(\tilde{x}_i)$ have alternating signs. Secondly, since $w_{n(i)}^\top x$ is a linear function in $x$, it can only changes sign for at most one time. This implies that $n(i) \neq n(j)$ if $j - i \geq 2$. Putting them together, we know that $n(i) \neq n(j)$ for $i \neq j$.

Recall that for each $1 \leq i \leq R - 2$, we have $(w_{n(i)}^\top x_a)(w_{n(i)}^\top x_b) < 0$. Therefore, there exists $R - 2$ neurons that are activated for either $x_a$ or $x_b$. This gives $N(x_a, W) + N(x_b, W) \geq R - 2$, which completes the proof.

$\square$

**Lemma 4.** *The sharpness of a neural network is lower-bounded by the number of active neurons:* $\lambda_{\max}\left(\nabla_W^2 L(W)\right) \geq \frac{r^2}{N^2} \sum_{i=1}^N N(x_i, W)$.

*Proof.* The Hessian of $l(W, x, y)$ can be expressed as

$$\nabla_W^2 l(W, x, y) = \begin{bmatrix} v_1 v_1^\top & \cdots & v_1 v_p^\top \\ \vdots & & \vdots \\ v_p v_1^\top & \cdots & v_p v_p^\top \end{bmatrix},$$

where $v_i = a_i \sigma'(w_i^\top x)x$. Suppose $V = [v_1^\top, \cdots, v_p^\top]$. As the nonzero eigenvalue of $V^\top V$ and $VV^\top$ is the same, this implies that

$$\lambda_{\max}(\nabla_W^2 l(W, x, y)) = \lambda_{\max}(VV^\top) = \sum_{i=1}^{p} \|v_i\|_2^2 = \sum_{i=1}^{p} \sigma'(w_i^\top x)\|x\|^2 \geq \sum_{i=1}^{p} \sigma'(w_i^\top x)r^2.$$

From the definition of $\nabla_W^2 L(W)$ and the positive definiteness of Hessian matrices, we know that

$$\lambda_{\max}\left(\nabla_W^2 L(W)\right) = \frac{1}{N}\lambda_{\max}\left(\sum_{i=1}^{N} \nabla_W^2 l(W, x_i, y_i)\right) \geq \frac{1}{N^2}\sum_{i=1}^{N} \lambda_{\max}\left(\nabla_W^2 l(W, x_i, y_i)\right).$$

Plug in the previous calculation and use the definition of $N(x)$, we have

$$\lambda_{\max}\left(\nabla_W^2 L(W)\right) \geq \frac{r^2}{N^2}\sum_{i=1}^{N}\sum_{j=1}^{p} \sigma'(w_i^\top x_j) = \frac{r^2}{N^2}\sum_{i=1}^{N} N(x_i, W).$$

$\square$

*Proof of Theorem 1.* Theorem 1 is a direct consequence of Assumption 2 and the following two lemmas.

$$\mathbb{E}_{X,X'}[R(X, X', W_t)] = \frac{1}{N^2}\sum_{i=1}^{N}\sum_{j=1}^{N} R(x_1, x_2, W_t)$$

$$\leq \frac{1}{N^2}\sum_{i=1}^{N}\sum_{j=1}^{N}(N(x_i, W_t) + N(x_j, W_t) + 2)$$

$$= \frac{2}{N}\sum_{i=1}^{N}(N(x_i, W_t) + 1)$$

$$\leq \frac{2N}{r^2}\lambda_{\max}(\nabla_W^2 L(W_t)) + 2$$

$$= O\left(\frac{N}{r^2\eta}\right)$$

$\square$

