# OpenReview forum: "Understanding Nonlinear Implicit Bias via Region Counts in Input Space"
_ICLR.cc/2025/Conference — Submitted to ICLR 2025_

### Official Review · Reviewer_V8HX · 2024-10-31

**Soundness:** 3
**Presentation:** 3
**Contribution:** 2
**Rating:** 6
**Confidence:** 3

**Summary:**

In this paper, the authors introduce new insights of understanding implicit bias of non-linear neural networks. They find that region count, a measure of the complexity of decision boundaries, is well correlated with generalization gap, defined as the difference between training and test errors. Their empirical and theoretical analysis show that the phenomenon of appropriate hyper-parameters leading to better generalization can be explained in terms of region counts. This work encourages future work for expanding their finding on more general and practical environments, further revealing the underlying implicit bias of neural networks.

**Strengths:**

- This paper introduces a novel and useful metric for understanding implicit bias of neural network’s generalization.
- Region count shows not only a high correlation with the generalization gap but also robustness across diverse architectures, datasets, and optimizers.
- This paper is well structured and well written. The authors’ claims are strongly supported by both theoretical analysis and empirical results on practical datasets.

**Weaknesses:**

- Proposed region count method assumes low-dimensional subspace of the training data. Although robustness to different dimension choices is shown in Section 7, it still relies on a substantial number of sampling trials (100 runs). However, Table 5 in Appendix B contains an ablation study on the impact of the number of experiment runs, which never been discussed in the main paper. Including these results in the main text would strengthen the method's validity.
- The proposed method provides an interesting explanation for the implicit bias whereby a large learning rate or small batch size facilitates superior generalization. While the authors state that this is 'typically deemed as beneficial for generalization’, but it is necessary to show that a large learning rate or small batch size leads to better generalization in terms of both generalization gap, and top-1 accuracy.
- Figure 4 and Table 8 report the correlation between generalization gap and region count on CIFAR-10 across different architectures. However, the range of generalization gap (y-axis) appears large compared to typical CIFAR-10 performance. For example, [1] reported that ResNet20 achieves an error rate of 8.75% on CIFAR-10, and [2] shows that ResNet18 achieves 93.02% accuracy, while in Table 8, ResNet18’s generalization gap is shown to be at least 15% (maximum 85% accuracy). Providing the top-1 accuracy for each architecture (under different hyper-parameters) would help prevent reader confusion.

[1] He, Kaiming, et al. "Deep residual learning for image recognition." *Proceedings of the IEEE conference on computer vision and pattern recognition*. 2016.

[2] https://github.com/kuangliu/pytorch-cifar

**Questions:**

- Does region count maintain a consistent correlation after dimensionality reduction (e.g., PCA)?
- Dose the variation in the dimensionality $\mathbb{R}^{d}$ affects the results of sampled subspace in region count?
- Is this paper first to understand the correlation between generalization gap and hyper-parameters?
- In Definition 1, how is 'Connectedness' validated in practice for any $f(\gamma(t))=c$ where$\ t \in [0,1]$? For example, is it achieved through a grid search?
- In Figure 7, it is unclear how to interpret the results of the random flip. Is the results missing?
- In Appendix B, the third paragraph mentions Table 4 twice.

---

> ### Author Response · Authors · 2024-11-17
> **Comments by Authors**
>
> We extend our gratitude to the reviewer for the comments and suggestions. Below, we address the primary concerns that have been raised.
>
> >Q1: Including Table 5 in Appendix B to the main text.
>
> **A1:** Thank you for the suggestion. In the next version, we will move Table 5 from Appendix B to the main text to highlight the robustness of our conclusions to the number of sampling trials, further validating the method’s reliability.
>
> >Q2: It is necessary to show that a large learning rate or small batch size leads to better generalization.
>
> **A2:** Extensive prior research [1][2][3] has demonstrated that large learning rates and small batch sizes typically lead to better generalization, both in terms of generalization gap and top-1 accuracy. However, this is not the primary focus of our paper. Our study aims to explore the relationship between model generalization and region count, providing a novel perspective that complements existing findings.
>
> >Q3: The range of generalization gap appears large compared to typical CIFAR-10 performance.
>
> **A3:** We appreciate the reviewer’s observation. To clarify, **data augmentation is not used during the training of the neural networks in our study**, whereas [4][5] applied data augmentation when training ResNet models, likely contributing to their higher test accuracy. In Section 7, we **also conduct experiments with data augmentation**. As shown in Figure 7, with optimal hyperparameters and data augmentation, the test accuracy reaches approximately 93%. Importantly, our correlation between region count and generalization gap **remains consistent** regardless of whether data augmentation is applied.
>
> >Q4: Does region count maintain a consistent correlation after PCA?
>
> **A4:** We thank the reviewer for suggesting this direction. In our paper, the subspace is generated using convex combinations of training data, allowing each point on the subspace to **be expressed as convex combinations of the training datapoint coordinates**. These coordinates can then be fed into the neural network for label prediction, enabling us to compute the region count based on the classification of labels across the subspace.
>
> However, when using PCA to generate a subspace, we can determine the coordinates of points within the reduced-dimensional space but **cannot map these points back to their original coordinates in the input space**. Without the original coordinates, it is **not possible to input these points into the neural network for label prediction**. Therefore, we have not yet identified a direct method for calculating the region count on a PCA subspace.
>
> >Q5: Dose the variation in the dimensionality $R^d$ affects the results of sampled subspace in region count?
>
> **A5:** The region count increases with subspace dimensionality. However, as shown in Table 2, the correlation between region count and generalization remains consistently high, unaffected by dimensionality changes.
>
> >Q6: Is this paper first to understand the correlation between generalization gap and hyper-parameters?
>
> **A6:** This paper does not directly analyze the correlation between generalization gap and hyperparameters. Instead, it studies the relationship between the generalization gap and region count, as well as region count and hyperparameters.
>
> >Q7: How is 'Connectedness' validated in practice?
>
> **A7:** Algorithm 1 in Appendix B details the computation of region count. The input space is divided into smaller grids, and **Breadth-First Search (BFS) is used to traverse adjacent grids with the same label**, identifying and counting connected components.
>
> >Q8: It is unclear how to interpret the results of the random flip.
>
> **A8:** Figures 6 and 7 present observations rather than interpretations, showing that mixup and data augmentation affect region count in different ways. These observations suggest that the mechanisms through which mixup and data augmentation improve model generalization may differ. Further exploration of this phenomenon is a promising direction for future research.
>
> >Q9: In Appendix B, the third paragraph mentions Table 4 twice.
>
> **A9:** Thank you for catching this typo. It has been corrected in the revised version.
>
> We thank the reviewer once again for the valuable and helpful suggestions.
>
> **References**
>
> [1] Keskar, Nitish Shirish, et al. "On large-batch training for deep learning: Generalization gap and sharp minima." arXiv preprint arXiv:1609.04836 (2016).
>
> [2] Jastrzębski, Stanisław, et al. "Three factors influencing minima in sgd." arXiv preprint arXiv:1711.04623 (2017).
>
> [3] Hoffer, Elad, Itay Hubara, and Daniel Soudry. "Train longer, generalize better: closing the generalization gap in large batch training of neural networks." Advances in neural information processing systems 30 (2017).
>
> [4] He, Kaiming, et al. "Deep residual learning for image recognition." Proceedings of the IEEE conference on computer vision and pattern recognition. 2016.
>
> [5] https://github.com/kuangliu/pytorch-cifar

---

> > ### Comment · Reviewer_V8HX · 2024-11-21
> > **Thank you for responding to my review**
> >
> > I appreciate the authors' efforts to address my concerns. Most of my questions have been resolved.
> >
> > However, one part remains unclear to me, though I don't think it's critical:
> >
> > - Q8) I'm still unclear about the observations of 'random flip'. The caption for Figure 7 states that it includes the impact of both 'random crop' and 'random flip'. However, the figures only show observations for 'random crop' (Correlation of Random Crop: 0.96, Impact of Random Crop). Please let me know if I have misunderstood something here.

---

> > > ### Author Response · Authors · 2024-11-21
> > > **The caption for Figure 7**
> > >
> > > We sincerely thank the reviewer for the feedback. In our experiments, we applied both random crop and random flip techniques for data augmentation and calculated the region count and generalization error. Figure 7 reflects the effect of applying both techniques. However, due to limited space in the figure title, we mentioned only "random crop" and omitted "flip." We apologize for the misunderstanding and have clarified this issue in the updated version of our paper. Please refer to the latest PDF for details.

---

> ### Comment · Reviewer_V8HX · 2024-11-29
>
> Thank you to the authors for their efforts. Based on their responses, I have decided to maintain my score. I hope the authors can incorporate these additional points into the final version of the paper.

---

> > ### Author Response · Authors · 2024-11-29
> > **Thank you!**
> >
> > We thank the reviewer for acknowledging our work! We will incorporate these additional points into the final version of the paper.

---

### Official Review · Reviewer_LVqU · 2024-11-03

**Soundness:** 2
**Presentation:** 2
**Contribution:** 1
**Rating:** 3
**Confidence:** 5

**Summary:**

The paper proposes to quantify implicit bias through the lens of linear regions in deep networks. The authors find some empirical relations between the number of linear regions and the ability of the architecture to generalize. Trends with respect to number of regions and hyper parameters are also highlighted.

**Strengths:**

The paper proposes to study an important problem which is the quantification of implicit bias with deep networks. While numerous measures have emerged, it does seems that using linear regions offers a promising direction. The paper also explores a few different explorative set of experiments showing trends with hyper parameters which can be useful to help decide about optimization or regularization settings to employ.

**Weaknesses:**

The major weakness of the paper is in failing to cite entire corpus of work that already explored that direction before.

Starting with the seminal work of Montufar (On the Number of Linear Regions of Deep Neural Networks) which also studies the impact of number of linear regions on performances and its tie with the architecture. Many follow up works by the same authors delve deeper into that exact research problem as well: Sharp bounds for the number of regions of maxout networks and vertices of minkowski sums. A convolutional network specific study also comes form (On the Number of Linear Regions of Convolutional Neural Networks).

In parallel a whole set of studies from Baraniuk also look at that exact problem:
- A Spline Theory of Deep Networks
- SplineCam: Exact Visualization and Characterization of Deep Network Geometry and Decision Boundaries
- Deep Networks Always Grok and Here is Why
all involving counting regions and relating that to test performance and generalization, as well as depicting comparisons with optimizers and architectures, as provided by the current submissions.

**Questions:**

Without citing prior work that look at the exact same problem studied here, and thus without any discussion or comparisons, it is hard to assess the novelty of the current submission. A priori, it seems that the proposed findings and methods have already been studied before hence making the current submission fall below acceptance level. However I encourage the authors to precisely cite and compare those references to their submissions and specifically demonstrate how/where do they provide novelty.

---

> ### Author Response · Authors · 2024-11-17
> **Comments by Authors**
>
> We thank the reviewer for the comments and constructive suggestions. In the following, we address the main concern raised. Please find the details below.
>
> >Q1: Starting with the seminal work of Montufar which also studies the impact of number of linear regions on performances and its tie with the architecture. Discuss the difference.
>
> **A1:** In the "Related Work" section, specifically under "Region Counts of Neural Networks," we discuss prior works on linear regions and have already clarified the distinction between linear regions and our definition of region count.
>
> Linear regions, as defined in prior works [1][2][3][4][5], refer to the set of inputs **corresponding to the same activation pattern** in the network. In contrast, our definition of decision regions refers to **connected areas in the input space that correspond to the same label**. Unlike linear regions, which are independent of the labels output by the neural network and primarily focus on the network's representation capability, our approach is more closely tied to the network's generalization ability. Therefore, the analysis of linear regions fundamentally differs from our work.
>
>
> >Q2: Discuss the relationship with the work of [6][7][8].
>
>
> **A2:** We thank the reviewer for suggesting these papers [6][7][8], and we have carefully read them.
>
> Paper [6] introduces a theoretical framework representing deep networks as max-affine spline operators (MASOs) via spline functions and operators. It proposes a regularization term based on this theory, improving classification performance. Paper [7] uses spline partition geometry to characterize and analyze the geometry of neural network decision boundaries, while paper [8] investigates training dynamics, such as grokking and delayed robustness, through the number of spline regions.
>
> These works provide an interesting perspective on partitioning the input space of neural networks using geometric properties, specifically leveraging the continuous piecewise linear (CPWL) activation functions. Their method **divides the input space into linear regions determined by hyperplanes derived from each network layer**, reflecting the geometric structure of the network rather than directly correlating with its output labels.
>
> In contrast, our method **partitions the input space based explicitly on the network's predicted labels**, ensuring that each region corresponds to a specific label. This distinction ties our approach more directly to the network’s labeling complexity and allows us to focus on its correlation with generalization ability. We will incorporate these points into the discussion of related work in the revised paper.
>
> Finally, we thank the reviewer once again for the effort in providing us with valuable suggestions. We will continue to provide clarifications if the reviewer has any further questions.
>
>
> **References**
>
> [1] Montufar, Guido F., et al. "On the number of linear regions of deep neural networks." Advances in neural information processing systems 27 (2014).
>
> [2] Xiong, Huan, et al. "On the number of linear regions of convolutional neural networks." International Conference on Machine Learning. PMLR, 2020.
>
> [3] Hanin, Boris, and David Rolnick. "Deep relu networks have surprisingly few activation patterns." Advances in neural information processing systems 32 (2019).
>
> [4] Hanin, Boris, and David Rolnick. "Complexity of linear regions in deep networks." International Conference on Machine Learning. PMLR, 2019.
>
>
> [5] Serra, Thiago, Christian Tjandraatmadja, and Srikumar Ramalingam. "Bounding and counting linear regions of deep neural networks." International conference on machine learning. PMLR, 2018.
>
> [6] Balestriero, Randall. "A spline theory of deep learning." International Conference on Machine Learning. PMLR, 2018.
>
> [7] Humayun, Ahmed Imtiaz, et al. "Splinecam: Exact visualization and characterization of deep network geometry and decision boundaries." Proceedings of the IEEE/CVF Conference on Computer Vision and Pattern Recognition. 2023.
>
> [8] Humayun, Ahmed Imtiaz, Randall Balestriero, and Richard Baraniuk. "Deep networks always grok and here is why." arXiv preprint arXiv:2402.15555 (2024).

---

> > ### Comment · Reviewer_LVqU · 2024-11-28
> > **Questions**
> >
> > I thank the authors for their answer, however I disagree with the authors comments that there is no connection between the two definitions (input space linear regions and class regions). First, at deeper layers, the inputs of similar classes will fall within the same linear regions of those deeper layers (hence making the two definition equivalent). This is happening as ultimately (last layer) if this isn't the case then separation can not occur. Second, there has been known positive relationship between the ability to have high count of input linear regions around training points and the ability of a model to classify those samples. This--to me--deserves at least a strong discussion and comparison section. See for example figure 3 of [1]. I am thus keeping my score.
> >
> > [1] NEURAL ARCHITECTURE SEARCH ON IMAGENET IN FOUR GPU HOURS:
> > A THEORETICALLY INSPIRED PERSPECTIVE
> > Wuyang Chen, Xinyu Gong, Zhangyang Wang

---

> > > ### Author Response · Authors · 2024-11-29
> > >
> > > We sincerely appreciate the reviewer for the feedback to our paper. We would like to clarify several points:
> > >
> > > First, we agree that **the number of linear regions provides an upper bound for the number of decision regions**, as points with identical activation patterns will fall within the same linear region and thus share the same output. However, the reverse is not necessarily true: **points with slightly different activation patterns may still yield the same label prediction, particularly in classification tasks where the output label space is limited**, despite the network’s potentially complex representations. This results in a significant difference in magnitude between the two quantities. For example, Figure 3 in [1] and Figure 4 in our paper **demonstrate a difference of roughly 1000 times**.
> > >
> > > Second, regarding the correlation between linear regions and test accuracy, we would like to emphasize that the points in Figure 3 of [1] **correspond to different network architectures**, where the training hyperparameters (learning rate, weight decay, batch size) are fixed. This analysis primarily investigates the relationship between the network's representation ability and its test accuracy, rather than how specific training strategies can improve performance. While it is intuitive that increasing a network's expressive capacity could allow it to represent more information and improve accuracy, it can also lead to over-parameterization, which helps explain the relatively weak correlation (approximately 0.5) observed in [1]. In contrast, our paper focuses on **a fixed network architecture, where each point in Figure 4 represents a different combination of training hyperparameters** (as shown in Table 1). Here, we aim to explore how training strategies influence generalization. Our proposed measure, region count, **is strongly correlated with the generalization gap** (approximately 0.98), offering a potential way to describe implicit bias and a promising tool for selecting hyperparameters that optimize generalization.
> > >
> > > We hope this clarifies our approach and highlights the distinction between the two analyses. Once again, we appreciate the reviewer’s thoughtful feedback. We would be happy to provide further clarifications if the reviewer has any additional questions.
> > >
> > >
> > > **References**
> > >
> > > [1] Chen, Wuyang, Xinyu Gong, and Zhangyang Wang. "Neural architecture search on imagenet in four gpu hours: A theoretically inspired perspective." arXiv preprint arXiv:2102.11535 (2021).

---

> ### Author Response · Authors · 2024-12-01
>
> Thanks again for your valuable feedback! Could you please let us know whether your concerns have been addressed? We are happy to make further updates if you have any other questions or suggestions.

---

### Official Review · Reviewer_SnZS · 2024-11-03

**Soundness:** 3
**Presentation:** 3
**Contribution:** 2
**Rating:** 5
**Confidence:** 3

**Summary:**

This paper introduces *region count* as a new metric for understanding generalization in neural networks by examining the consistency of predictions across connected regions in the input space. The authors suggest that region count, which reflects the complexity of a model’s decision boundaries, is a more effective generalization indicator than traditional parameter-based metrics. Empirical results across various architectures and hyperparameter settings show that models with lower region counts tend to generalize better. Additionally, the authors' theoretical analysis links large learning rates to simpler decision boundaries, which may enhance generalization.

**Strengths:**

1. **Innovative Metric**: The introduction of region count to measure implicit bias in non-linear neural networks addresses limitations in parameter-dependent metrics, offering a fresh perspective on generalization.
3. **Hyperparameter Insights**: Findings around learning rate and batch size impacting region count offer practical value, guiding hyperparameter choices for improved generalization.
4. **Theoretical Basis**: Theoretical analysis, albeit in a simplified setup, provides a foundation for understanding the implicit bias induced by large learning rates, which aligns with the empirical findings.

**Weaknesses:**

### Weaknesses

1. **Limited Scope of Theoretical Analysis**: The theoretical section mainly focuses on a two-layer ReLU model, which limits its generalizability. Extending this analysis to more complex architectures would strengthen the theoretical contributions.

2. **Scalability Challenges in High Dimensions**: Estimating region count in high-dimensional spaces can be computationally intensive. While the proposed low-dimensional approximation is valuable, further analysis of its scalability is needed.

3. **Narrow Experimental Domain**: The experiments are primarily conducted in the image domain, which restricts the broader applicability of the region count metric. Extending validation to tasks like NLP would enhance the metric’s relevance across varied contexts.

4. **Limited Comparative Analysis with Complexity Metrics**: Although norm-based metrics are discussed, deeper comparisons with other complexity measures—such as sharpness- and margin-based metrics—would offer a more comprehensive view of the advantages and limitations of region count.

The work of Andriushchenko et al. (2023) highlights how sharpness correlates positively with ResNet architectures but is less effective on Transformers and ViTs. For example, CIFAR and ResNet benchmarks alone may not sufficiently clarify the relationship between sharpness and generalization. Similarly, flatness measures perform well on CNNs, a setting similar to the one used here. Extending the method to NLP datasets and Transformer architectures, however, could better demonstrate its broader utility.

The empirical study’s impact is limited because the evaluated networks belong to the same family, raising questions about scalability to modern architectures like Transformers and ViTs. Andriushchenko’s findings also indicate that other flatness metrics struggle to adapt to these modern settings, suggesting that additional validation on ResNet and Transformer models could yield stronger insights. Though the method is claimed to be effective, the evidence provided lacks sufficient experimental depth. A more robust evaluation could include comparisons with sharpness metrics where they perform well or tests on edge cases, such as out-of-distribution generalization, to better establish the method’s efficacy.

**Questions:**

1. **Link to Certified Radius and Generalization Measures**: The connection between certified radius, margin, and norm-based measures may not be sufficient. Could metrics such as the ratio of a neural network’s margin to its Lipschitz constant provide additional insight? This ratio is directly related to the certified radius, representing the region where a classifier’s predictions remain unchanged (see Tsuzuku et al., 2018).

2. **Scalability**: How does the computational cost of estimating region count scale with model size and dimensionality, especially in NLP domains?

3. **Extensions to Complex Architectures**: Can the theoretical framework be feasibly extended to deeper or more sophisticated architectures, and would insights about learning rate and region count remain applicable?

4. **Generalization Beyond Classification**: Could region count metrics be adapted to other types of tasks, such as regression or structured prediction?

5. **Impact of Additional Hyperparameters**: Beyond learning rate and batch size, how do other hyperparameters—like weight decay, optimizer choice, and normalization layers such as BatchNorm or LayerNorm—affect region count and generalization?

6. **Comparison with Other Generalization Metrics**: How does the generalization potential of region counts compare with other metrics such as Lipschitz continuity, flatness, etc.? (See Jiang et al., 2020).

7. **Role of Implicit Regularization**: High learning rates (linked to weight decay effects with normalization) and small batch sizes (as noted by Keskar et al., 2017) are already associated with implicit regularization. The paper suggests that this is explained through sharp minima, but a more precise explanation could clarify this link.

8. **Connection to Certified Radius in Adversarial Robustness**: Discuss the relationship between region count and certified radius, particularly regarding Tsuzuku’s work, which uses the ratio of margin and Lipschitz constant as a robustness measure.

9. **Comparison to Sharpness**: Sharpness measures are data-dependent, yet the proposed method is also data-dependent. Would a comparison with sharpness metrics provide valuable insights?

10. **Scalability to Larger Models**: Does the slicing method scale effectively to larger models, and how does it behave as the number of parameters increases? Testing it on MLP layers with an increasing number of features could offer insights.

11. **Effectiveness of Scaling Techniques**: The scaling technique to invalidate the measure does not seem effective for the margin-to-Lipschitz-based ratio. Are there alternative approaches to validate or refine this measure?

12. **Region Count Interpretation**: Could you clarify what is meant by region count when it is not an integer?

### References

- Tsuzuku, Y., Sato, I., & Sugiyama, M. (2018). *Lipschitz-Margin Training: Scalable Certification of Perturbation Invariance for Deep Neural Networks*. In Proceedings of the 32nd Conference on Neural Information Processing Systems (NeurIPS).

- Jiang, Y., Neyshabur, B., Mobahi, H., Krishnan, D., & Bengio, S. (2020). *Fantastic Generalization Measures and Where to Find Them*. In Proceedings of the 8th International Conference on Learning Representations (ICLR).

- Keskar, N. S., Mudigere, D., Nocedal, J., Smelyanskiy, M., & Tang, P. T. P. (2017). *On Large-Batch Training for Deep Learning: Generalization Gap and Sharp Minima*. In Proceedings of the 34th International Conference on Machine Learning (ICML).

---

> ### Author Response · Authors · 2024-11-17
> **Comments by Authors**
>
> We thank the reviewer for the time spent on reviewing our work and for the very detailed comments. We **add experiments comparing against more metrics such as sharpness-based and Pac-Bayesian based**. Although our work aims to identify an interpretable implicit bias for nonlinear models rather than predicting the generalization gap, we are happy to add more empirical results if that could help strengthen our findings. Please find the details below.
>
> >Q1: Link to Certified Radius [5] and Generalization Measures.
>
> **A1:** We have carefully reviewed the paper on Certified Radius [5]. The margin-to-Lipschitz constant proposed in these works is primarily designed to **measure a neural network's robustness against adversarial perturbations, rather than its generalization ability**. Adversarial robustness aims to reduce vulnerability to intentionally crafted small input perturbations, while generalization focuses on the network's performance on unseen data. Since the input distributions in these two scenarios differ significantly, there is no direct connection between the two concepts.
>
> >Q2: How does the computational cost of estimating region count scale with model size and dimensionality, especially in NLP domains?
>
> **A2:** In Appendix B, we outline the method for calculating region count, which has a time complexity that does not scale with model size but instead grows exponentially with the dimension of the selected subspace. Our paper focuses on classification tasks, where regions are defined based on label consistency. Extending the definition of regions to NLP tasks remains an open challenge.
>
> To explore the applicability of our approach to transformer-based models, we conduct an experiment using Vision Transformers (ViT) on CIFAR-10. The results show **a strong correlation of 0.84** between region count and the generalization gap, reinforcing the validity of our measure in this setting. Details of the hyperparameters used are summarized below:
>
> |Hyperparameters|Value|
> |:---:|:----:|
> |Learning rate|1e-4,5e-5,1e-5|
> |Batch size|256,512,1024|
> |Weight decay|1e-5,1e-6,1e-7|
>
> >Q3: Can the theoretical framework be feasibly extended to deeper or more sophisticated architectures?
>
> **A3:** We appreciate the reviewer’s insightful question. Computing region count for more complex architectures is indeed a challenging task, as it cannot be directly bounded using activation numbers. At present, we have not identified a feasible method for theoretical analysis in these cases. We consider this an important avenue for future research.
>
> >Q4: Could region count metrics be adapted to other types of tasks, such as regression or structured prediction?
>
> **A4:** We thank the reviewer for this thoughtful suggestion. To adapt region count metrics to regression tasks, a key step would be to define connectedness. One potential approach is to partition the regression error into small intervals and analyze the connectivity of input space points falling within each interval. We believe this is a promising direction for future exploration.
>
> >Q5: Beyond learning rate and batch size, how do other hyperparameters—like weight decay, optimizer choice, and normalization layers affect region count and generalization?
>
> **A5:** We thank the reviewer for this insightful question. Our preliminary experiments indicate that the relationships between these hyperparameters and region count are highly complex, making it challenging to capture them systematically within the scope of the current work. For this reason, we did not include detailed analyses in the main text. We consider this an important area for future research.
>
> >Q6: How does the generalization potential of region counts compare with other metrics such as Lipschitz continuity and flatness?
>
> **A6:** We thank the reviewer for this question. To compare the effectiveness of region count with other metrics, we conduct additional experiments on CIFAR-10 using ResNet-18. We measured the correlation between the generalization gap and spectral norm [1], PAC-Bayesian flatness [2], and region count:
>
> |Measure|Spectral Norm|PAC-Bayesian Flatness|Region Count|
> |:---:|:----:|:----:|:----:|
> |Correlation|0.77|-0.31|**0.98**|
>
> The results demonstrate that region count exhibits a significantly stronger correlation with the generalization gap.
>
> >Q7: High learning rates (linked to weight decay effects with normalization) and small batch sizes are already associated with implicit regularization. The paper suggests that this is explained through sharp minima, but a more precise explanation could clarify this link.
>
> **A7:** Numerous prior studies [6][7][8] have investigated the effects of learning rate and batch size on generalization, showing that large learning rates and small batch sizes often lead to better generalization. We acknowledge that our explanation could be more precise, and we will include additional clarification and discussion on this topic in the next version of the paper.

---

> ### Author Response · Authors · 2024-11-17
> **Comments by Authors**
>
> >Q8: Discuss the relationship between region count and certified radius[5], which uses the ratio of margin and Lipschitz constant as a robustness measure.
>
> **A8:** We thank the reviewer for this insightful question. Region count and certified radius are indeed related, as both quantify changes in a neural network's predictions within the input space. The key difference lies in their scope: region count is **a global measure**, capturing the network's overall prediction behavior across the entire input space, while certified radius is **a local measure**, focusing on the extent of perturbation required to alter a prediction. A promising direction for future work could involve exploring whether certified radius provides an upper bound for region count.
>
> >Q9: Would a comparison with sharpness metrics provide valuable insights?
>
> **A9:** We thank the reviewer for the question. To address this, we conduct additional experiments on CIFAR-10 using ResNet-18, comparing sharpness metrics from PAC-Bayesian bounds, including those using the origin and initialization as reference tensors (PB-I and PB-O), as well as PAC-Bayesian Magnitude-aware Perturbation Bounds (PB-M-I and PB-M-O) [2][3][4]. The results are summarized below:
>
> |Measure|PB-I|PB-O|PB-M-I|PB-M-O|Region Count|
> |:---:|:----:|:----:|:----:|:----:|:----:|
> |Correlation|-0.35|-0.31|0.79|0.78|**0.98**|
>
> The results demonstrate that region count shows a significantly stronger correlation with the generalization gap compared to these sharpness metrics.
>
> >Q10: Does the slicing method scale effectively to larger models?
>
> **A10:**  In our paper, we already explored the correlation across neural networks with different parameter sizes, finding it to be largely consistent. To further address this question, we conduct additional experiments using **ResNet models with varying depths** on CIFAR-10. The results are summarized below:
>
> |Network|Resnet18|Resnet34|Resnet50|Resnet101|Resnet152|
> |:---:|:----:|:----:|:----:|:----:|:----:|
> |Correlation|0.98|0.98|0.97|0.98|0.96|
>
> These results indicate that the correlation remains consistently high as the number of parameters increases.
>
> >Q11: The scaling technique to invalidate the measure does not seem effective for the margin-to-Lipschitz-based ratio. Are there alternative approaches to validate or refine this measure?
>
> **A11:** We thank the reviewer for raising this question. The Lipschitz constant of a neural network is upper-bounded by the product of the spectral norms of its weight matrices across all layers. To evaluate the effectiveness of the margin-to-Lipschitz-based ratio, we conduct experiments to verify the correlation between the sum of spectral norms over margin [4] and the generalization gap. The results are summarized below:
>
> |Measure|SPECTRAL/MARGIN|Region Count|
> |:---:|:----:|:----:|
> |Correlation|0.32|**0.98**|
>
> The results indicate that the margin-to-Lipschitz-based ratio shows a weak correlation, suggesting it may not be a reliable predictor of generalization compared to region count.
>
> >Q12: Could you clarify what is meant by region count when it is not an integer?
>
> **A12:** Region count is defined **as an expectation**. As detailed in Appendix B, we calculate it by randomly sampling 100 subspaces and averaging the region counts across these subspaces. While the region count for each individual subspace is an integer, **the averaged value is not necessarily an integer**.
>
> We thank the reviewer once again for the valuable and helpful suggestions. We would be happy to provide further clarifications if the reviewer has any additional questions.
>
> [1] Bartlett, Peter L., Dylan J. Foster, and Matus J. Telgarsky. "Spectrally-normalized margin bounds for neural networks." Advances in neural information processing systems 30 (2017).
>
> [2] Keskar, Nitish Shirish, et al. "On large-batch training for deep learning: Generalization gap and sharp minima." arXiv preprint arXiv:1609.04836 (2016).
>
> [3] Neyshabur, Behnam, et al. "Exploring generalization in deep learning." Advances in neural information processing systems 30 (2017).
>
> [4] Jiang, Yiding, et al. "Fantastic generalization measures and where to find them." arXiv preprint arXiv:1912.02178 (2019).
>
> [5] Tsuzuku, Yusuke, Issei Sato, and Masashi Sugiyama. "Lipschitz-margin training: Scalable certification of perturbation invariance for deep neural networks." Advances in neural information processing systems 31 (2018).
>
> [6] Keskar, Nitish Shirish, et al. "On large-batch training for deep learning: Generalization gap and sharp minima." arXiv preprint arXiv:1609.04836 (2016).
>
> [7] Jastrzębski, Stanisław, et al. "Three factors influencing minima in sgd." arXiv preprint arXiv:1711.04623 (2017).
>
> [8] Hoffer, Elad, Itay Hubara, and Daniel Soudry. "Train longer, generalize better: closing the generalization gap in large batch training of neural networks." Advances in neural information processing systems 30 (2017).

---

> ### Author Response · Authors · 2024-12-01
>
> Thanks again for your valuable feedback! Could you please let us know whether your concerns have been addressed? We are happy to make further updates if you have any other questions or suggestions.

---

> > ### Comment · Reviewer_SnZS · 2024-12-02
> >
> > I thank the authors for their responses, but I believe they did not fully address my concern regarding the certified radius. Specifically, they considered the ratio as Lipschitz over margin, while the certified radius is typically defined as margin over Lipschitz. Additionally, it would have been beneficial to provide more theoretical insights into the proposed method.
> >
> > That said, the authors have addressed some of my other concerns. However, I still believe the paper should not be accepted. Nevertheless, based on their clarifications, I am upgrading my score to 5.

---

### Official Review · Reviewer_o6TY · 2024-11-04

**Soundness:** 3
**Presentation:** 3
**Contribution:** 3
**Rating:** 6
**Confidence:** 5

**Summary:**

This paper motivates and studies a novel generalization measure for neural networks: the number of connected regions in the input space. The paper first describes current challenges in connecting generalization to geometric properties of neural networks, and then introduces the proposed generalization measure. Extensive experiments assess the correlation between a small number of connected regions and generalization. Finally, the paper concludes by providing a theoretical link between the region count (in the training data) and the learning rate of (S)GD used during training.

Update: changed my score to 6.

**Strengths:**

This is an interesting paper to read. I would like to highlight the following strengths:

- Novel (empirical) measure for generalization: To the best of my knowledge, the proposed measure (input space connected region count) has not been studied theoretically or experimentally before, and the experimental findings are interesting. Namely, the paper finds that, for various configurations, the proposed measure (or, better, an estimate of it) seems to correlate well with the generalization gap.
- Extensive experiments: The experimental analysis is very thorough, including many datasets, models and hyperparameter choices.
- Theoretical result on the relation between region count and learning rate: Theorem 1 (partially) explains the empirical finding that a larger learning rate is associated with a smaller region count (and, as a result, smaller generalization gap). I found the result simple yet creative.

**Weaknesses:**

I believe that the community will find the findings and observations of this paper interesting. However, I identified a few areas where the paper could improve substantially:

- Omission of important details from introduction: The paper almost exclusively studies a specific approximation of the input space region count, which measures the number of prediction changes along *a line that connects two points from the training data*. However, this detail is not mentioned in the introduction, but only much later. Furthermore, many readers, while going through the paper for the first time, might confuse the introduced measure with the number of linear regions of the neural network. While this is clarified later in the paper (pg. 3), a short explanation in the introduction would be helpful.

- Weak/misleading section on "Motivation" (Section 3): I found Section 3 to be entirely misleading. The norm and margin quantities considered have **no reason** to be correlated with generalization for a ResNet18. First, they are not properly normalised, which is acknowledged later, so I do not see the reason for considering them in the first place. Second, there is no clear understanding that SGD on a ResNet will necessarily increase a specific notion of margin or minimise a specific norm. If someone wants to assess such connections, they should probably focus on simpler models (such as homogeneous neural networks) and control for many confounders. I understand and agree with the general point of this section (i.e., that accurate generalization measures based on the parameters of a "practical" network may be challenging to find in practice), but I found the motivating experiments problematic. I insist on this, since this section is the starting point of the paper and might mislead many readers. I would suggest being more precise in this section. Further concrete comments on this section: 1) Equation in line 172 is missing the distribution with respect to the expectation is taken. This is crucial, as it is not clear whether it applies to train or test data (it should be train). 2) It would be good to define the input-space margin which you mention in line 188 for the first time.

- Insufficient theoretical link to generalization: While I understand this is mainly an experimental paper, I was disappointed that there is no discussion on how the proposed measure can perhaps be related to improved generalization. For example, there is no mention of the self-evident property that a very small region count is undesirable (for region count equal to 1, we obtain a trivial predictor). An example of a satisfying result would be that (S)GD biases the model to implicitly minimise $R(\theta)$ under the constraint that all the train points are classified correctly together with perhaps more constraints from the hyperparameters (akin to results that exists for gradient descent on homogeneous neural networks and margin maximization). Should we hope to prove such a result? Do you believe that such result could be true? While this point alone is not brought up to dissuade acceptance of the paper, I would appreciate any thoughts the authors have on this.

**Questions:**

- line 362: why is there a norm in the definition of the $\ell_2$ loss? The neural network is defined to have real output.
- line 366: remove the word "the" before "assumption".
- line 381: the term "Hessian $\ell_2$ norm" sounds strange.
- Table 2: what are the reported results? correlation? This is not mentioned.
- There seems to be a typo in line 797 ("Table 4" appears twice).
- proof of Lemma 2, line 988: shouldn't the condition for the inner products be for $i$ and $i+1$, instead of $i$ and $i+2$?
- line 042, "results from linear regime can be extended to linear neural networks": this sentence is confusing. Similarly in line 037. Linear neural networks are in the linear regime. You can just mention that results can be extended for the deep case.

---

> ### Author Response · Authors · 2024-11-17
> **Comments by Authors**
>
> We greatly appreciate the reviewer's comments and valuable suggestions. We address the reviewer's questions in more detail as follows:
>
> >Q1: Omission of important details from introduction.
>
> **A1:** We appreciate the reviewer's feedback on the missing details. In the updated version of our paper, we have revised the introduction to include the definition of region count and its distinction from the concept of linear regions. The changes are highlighted in blue.
>
> >Q2: The section on 'Motivation' in Section 3 is misleading. There are also some typos in Section 3.
>
> **A2:** The motivation section emphasizes that commonly used measures for characterizing implicit bias, such as norm and margin, are valid in specific cases (e.g., linear or homogeneous networks) [1][2][3]. However, these measures **fail to perform well for more general nonlinear networks like ResNet-18**. Previous research [4] also shows that these measures do not correlate with the generalization gap. Our experiment further demonstrates their limitations, highlighting the need for a new approach to characterizing implicit bias in general neural networks. This led us to **introduce region count, which effectively captures implicit bias**. We have addressed the typos and revised Section 3 to improve clarity, with changes highlighted in blue.
>
>
> >Q3: Insufficient theoretical link to generalization.
>
> **A3:** Thank you for raising this question. The question and suggested theorem are more closely related to optimization rather than generalization. A classifier with a very small region count could indeed perform poorly. however, it **does not necessarily contradict the possibility of having a small generalization gap**. For instance, a naive classifier with region count 1 will perform poorly on both the training and test datasets, resulting in a small generalization gap.
>
> The suggested theorem could potentially be achieved by proving that large learning rates still allow convergence to a minimizer under the setup considered in Section 6.2. We agree with the reviewer that this paper primarily focuses on the empirical and generalization aspects. A more rigorous theoretical analysis of the tradeoff between generalization and optimization in relation to region counts is a valuable direction for future work, and we plan to explore this in subsequent research.
>
> >Q4: Minor typos.
>
> **A4:** We thank the reviewer for pointing out the typos. These have been corrected in the updated version, with changes highlighted in blue. Specifically, for line 988 in the proof of Lemma 2, the condition for the inner products involving $i$ and $i+2$ is not a typo. We have added a more detailed explanation in the appendix to clarify this point.
>
> Finally, we thank the reviewer once again for the effort in providing us with valuable and helpful suggestions. We will continue to provide clarifications if the reviewer has any further questions.
>
> **Reference**
>
> [1] Soudry, Daniel, et al. "The implicit bias of gradient descent on separable data." Journal of Machine Learning Research 19.70 (2018): 1-57.
>
> [2] Ji, Ziwei, and Matus Telgarsky. "The implicit bias of gradient descent on nonseparable data." Conference on learning theory. PMLR, 2019.
>
> [3] Lyu, Kaifeng, and Jian Li. "Gradient descent maximizes the margin of homogeneous neural networks." arXiv preprint arXiv:1906.05890 (2019).
>
> [4] Jiang, Yiding, et al. "Fantastic generalization measures and where to find them." arXiv preprint arXiv:1912.02178 (2019).

---

> ### Comment · Reviewer_o6TY · 2024-11-20
> **Reply**
>
> Thank you very much for your reply and for taking the time to answer my questions.
>
> Most of the changes in the manuscript look very good. Specifically,
>
> > Specifically, for line 988 in the proof of Lemma 2, the condition for the inner products involving  and  is not a typo. We have added a more detailed explanation in the appendix to clarify this point.
>
> I see. Thank you for explaining this further. There is a minor typo in the included equation: In lines 992 and 995, it should be $\tilde{x}$, instead of $x$.
>
> To be honest, my concerns with the very problematic Section 3 have not been addressed, since I find the train of thought very misleading. As I stated in my review, "the norm and margin quantities considered have no reason to be correlated with generalization for a ResNet18". They are generalization measures for a different learning problem. It is akin to using generalization guarantees for decision trees and claiming that the theory is wrong if they do not correlate with generalization in transformers with skip connections and relative positional encoding. However, since I agree with the high-level idea of this section and the authors do cite [Jiang et al., 2019], where a more thorough discussion is included, I will not insist on this further. I leave it to the readers to think critically. I updated my score to reflect this.

---

> > ### Author Response · Authors · 2024-11-21
> > **Reply**
> >
> > Thank you for your feedback and recognition of our efforts! We will further refine the discussion on motivation in Section 3 in the next version of the paper.

---

### Author Response · Authors · 2024-11-23

We would like to express our sincere gratitude for the reviewer's constructive suggestions and comments. Since the deadline is approaching, we sincerely hope the reviewers can read our response. Please let us know if the reviewers have any comments about our response or any other additional concerns. We are eager to provide any further clarifications and discussions to help the evaluation.

---

### Meta-Review · Area_Chair_FjWa · 2024-12-22

**Metareview:**

This paper proposes region count in input space as a measure of generalization. The paper empirically demonstrates the strong correlation between region count and generalization gap. The paper also theoretically shows that larger learning rate can lead to smaller region count. The experimental results are thorough and strong, and the theoretical result is novel and the connection to edge-of-stability is interesting. A key limitation of the work is that region count is computationally intractable in the high-dim space and the paper adopts an approximation of it in a very low dimension. While experiments show that the correlation remains robust across varying dimensions, the work lacks a principled justification for the reliability of this approximation.

**Additional Comments On Reviewer Discussion:**

Reviewer LVqU had a major concern about the lack of discussion/comparison between region count and linear regions. The authors explained their differences in the rebuttal, and the AC found the explanation sufficient.

---

### Decision · Program_Chairs · 2025-01-22

Reject